# Upregulation of Neuropilin-1 Inhibits HTLV-1 Infection

**DOI:** 10.3390/pathogens12060831

**Published:** 2023-06-15

**Authors:** Wesley Kendle, Kimson Hoang, Erica Korleski, Amanda R. Panfil, Nicholas Polakowski, Isabelle Lemasson

**Affiliations:** 1Department of Microbiology and Immunology, Brody School of Medicine, East Carolina University, Greenville, NC 27834, USA; kendlew17@students.ecu.edu (W.K.); hoangk@ecu.edu (K.H.); korleskierica@gmail.com (E.K.); polakowskin@ecu.edu (N.P.); 2Center for Retrovirus Research, Department of Veterinary Biosciences, College of Veterinary Medicine, The Ohio State University, Columbus, OH 43210, USA; panfil.6@osu.edu

**Keywords:** HTLV-1, Nrp1, HBZ, transcription, p300/CBP, bZIP, infection

## Abstract

Infection with human T-cell leukemia virus type 1 (HTLV-1) can produce a spectrum of pathological effects ranging from inflammatory disorders to leukemia. In vivo, HTLV-1 predominantly infects CD4^+^ T-cells. Infectious spread within this population involves the transfer of HTLV-1 virus particles from infected cells to target cells only upon cell-to-cell contact. The viral protein, HBZ, was found to enhance HTLV-1 infection through transcriptional activation of *ICAM1* and *MYOF*, two genes that facilitate viral infection. In this study, we found that HBZ upregulates the transcription of *COL4A1*, *GEM*, and *NRP1*. *COL4A1* and *GEM* are genes involved in viral infection, while *NRP1*, which encodes neuropilin 1 (Nrp1), serves as an HTLV-1 receptor on target cells but has no reported function on HTLV-1-infected cells. With a focus on Nrp1, cumulative results from chromatin immunoprecipitation assays and analyses of HBZ mutants support a model in which HBZ upregulates *NRP1* transcription by augmenting recruitment of Jun proteins to an enhancer downstream of the gene. Results from in vitro infection assays demonstrate that Nrp1 expressed on HTLV-1-infected cells inhibits viral infection. Nrp1 was found to be incorporated into HTLV-1 virions, and deletion of its ectodomain removed the inhibitory effect. These results suggest that inhibition of HTLV-1 infection by Nrp1 is caused by the ectodomain of Nrp1 extended from virus particles, which may inhibit the binding of virus particles to target cells. While HBZ has been found to enhance HTLV-1 infection using cell-based models, there may be certain circumstances in which activation of Nrp1 expression negatively impacts viral infection, which is discussed.

## 1. Introduction

Human T-cell leukemia virus type 1 (HTLV-1) is a complex retrovirus that primarily infects CD4^+^ T-cells in vivo. Worldwide, 5–10 million people are estimated to be infected with HTLV-1, of which 5–10% will experience pathological effects associated with the viral infection [1,2]. Specifically, HTLV-1 is the etiologic agent of an often fatal form of leukemia designated adult T-cell leukemia (ATL) and, separately, a progressive inflammatory neurodegenerative disease known as HTLV-1-associated myelopathy/tropical spastic paraparesis (HAM/TSP) [3,4]. HTLV-1 infection is additionally associated with other inflammatory maladies that include infective dermatitis, uveitis, polymyositis, and Sjogren’s syndrome [5].

Infectious spread of HTLV-1 within the T-cell population requires direct contact between HTLV-1-infected and target T-cells. Once cell-to-cell contact is established, virions are transferred to target cells through a virological synapse, cellular conduits, or from virions contained in an extracellular biofilm-like matrix that is released from the surface of the infected cell [6,7,8,9]. The specific details of each of these infection mechanisms are highly complex (reviewed in [9]) and are not believed to be mutually exclusive. Subsequently, HTLV-1 virions bind to the cell surface and fuse with the plasma membrane via interactions of the surface unit (SU) of the HTLV-1 envelope protein with three receptors: heparin sulfate proteoglycans (HSPGs), neuropilin 1 (Nrp1), and glucose transporter 1 (Glut1) [10,11,12]. These receptors are believed to act in concert as a tri-receptor complex with HSPGs mediating initial virion attachment that, through interactions between HSPG and Nrp1, deliver the virion to Nrp1 to establish high-affinity binding. At this stage, a conformational change in SU is believed to promote Glut1 binding, which induces fusion and entry [13].

Nrp1 displays a diverse array of coreceptor functions. It interacts with plexins to mediate class 3 Semaphorin signaling, which is involved in repulsive axon guidance [14,15]. It also functions as a coreceptor for multiple growth factor receptors involved in angiogenesis, specifically augmenting signaling activated by vascular endothelial growth factor 165, platelet-derived growth factor-B, hepatocyte growth factor, and fibroblast growth factor [16,17,18,19]. In addition, Nrp1 enhances signaling through transforming growth factor β1 (TGF-β1; [20]), which is associated with maintaining regulatory T-cells and, through a non-canonical signaling pathway, has also been implicated in cancer progression [21]. The composition of the extracellular region of Nrp1 is critical to the binding of this diverse set of signaling ligands. This region of the protein contains two tandem N-terminal CUB domains (a1 and a2) for Semaphorin binding followed by two tandem Factor V/VIII homology domains (b1 and b2) for growth factor binding followed by a membrane-proximal MAM domain (c) that is proposed to position the other domains away from the membrane, allowing for an extended ectodomain [22,23].

The HTLV-1 protein, HTLV-1 basic leucine zipper (bZIP) factor (HBZ), regulates transcription using its capacity to interact with an array of cellular transcriptional regulators. HBZ contains an N-terminal activation domain with two LxxLL motifs that mediate high-affinity binding to the paralogous cellular coactivators p300 and CBP (interchangeably denoted p300/CBP) [24,25]. Within its C-terminal region, HBZ contains a leucine zipper (ZIP) domain that forms heterodimers with certain cellular bZIP factors, including Jun proteins, members of the Maf family, and certain members of the ATF/CREB family [26,27,28,29,30,31,32,33,34]. In addition, HBZ has been reported to interact with other transcriptional regulators [35,36]. One consequence of its transcriptional regulator function is to enhance HTLV-1 infection through activation of two cellular genes important for this process. Specifically, HBZ induces the expression of myoferlin, which abrogates lysosomal-mediated degradation of the HTLV-1 envelope protein and promotes cell adhesion [37]. In addition, HBZ upregulates the expression of ICAM-1 [38], which facilitates the binding of infected cells to target cells and promotes the formation of the virological synapse [9,39]. The expression of ICAM-1 is also activated by the HTLV-1 protein, Tax, which plays an essential role in viral infection [9]. From the stimulation of NF-κB signaling and by directly acting as a transcription factor, Tax activates the expression of several additional host genes involved in HTLV-1 infection [9]. Therefore, in HTLV-1-infected T-cells, HBZ may augment viral infection efficiency that is critically mediated by Tax.

In this study, we provide evidence that HBZ increases the expression of two other genes that contribute to HTLV-1 infection: *COL4A1* and *GEM*. Additionally, we found that HBZ increases the expression of *NRP1*. While contributions of the former genes to viral infection have been investigated [40,41], no role for Nrp1 expressed on HTLV-1-infected cells has been reported, which prompted us to investigate Nrp1 further. We identified an enhancer downstream of the *NRP1* gene that was bound by HBZ, resulting in increased recruitment of Jun proteins and p300/CBP to this chromosomal region. Analysis of HBZ mutants and results from ChIP assays support a primary role for HBZ in increasing the binding of Jun proteins to the enhancer. Unexpectedly, mutations in the LxxLL motifs of HBZ did not reduce *NRP1* transcription, indicating that HBZ is not directly involved in the recruitment of p300/CBP to this enhancer. In in vitro infection assays, Nrp1 inhibited infection, an effect that was associated with its incorporation into cell-free virions produced by HTLV-1-infected T-cell lines. Finally, using HEK293T cells, we found that inhibition of infection by Nrp1 was abolished by the deletion of its extracellular domain, suggesting that the extended ectodomain of Nrp1 on HTLV-1 virions inhibits infection. Given the overall positive role of HBZ toward viral infection, we speculate that negative effects on infection caused by the activation of Nrp1 might only arise during specific stages of HTLV-1 pathogenesis, such as progression from an indolent to an aggressive form of ATL, which is discussed. 

## 2. Materials and Methods

### 2.1. Plasmids 

pSG-Tax, pSG-Tax-His, pCMVHT1, pCRU5HT1-inLuc, and pHCMV-G have been described [38,42,43,44,45]. pLJM1 was a gift from Joshua Mendell (Addgene plasmid # 91980) [46]. Nrp1 expression vectors were generated by cloning DYK-tagged NRP1 from pcDNA3.1-C-(k)-NRP1-DYK (GenScript) into pQCXIP (Clontech) and pLJM1 at the BamH1 and EcoRI sites, respectively, using a Gibson Assembly Cloning Kit (New England Biolabs Inc., Ipswich, MA, USA). shRNA vectors shGFP (SHC202) and shNRP1 [TRCN0000300917 (MT-2 transductions), TRCN0000322980 (Jurkat transfections and ATL-2 transductions)] were purchased from MilliporeSigma. pQCXIP-NRP1-TM was constructed via PCR amplification of the transmembrane/cytoplasmic domain sequence, which was inserted into the PacI and EcoRI sites. pQCXIP-NRP1-Δabc was constructed via amplification of the signal peptide sequence, which was inserted into the BglII and MluI sites of pQCXIP-NRP1-TM (an MluI site had been added to the forward primer used to amplify the NRP-1-TM sequence).

### 2.2. Cell Culture 

Jurkat, CEM, C8166/45, MT-2, SLB-1, and ATL-2 cells were cultured in Isocove’s modified Dulbecco medium (IMDM). Primary CD4^+^ lymphocytes, TL-Om1, and HTLV-1-immortalized lymphocyte cell lines [47] were cultured in Roswell Park Memorial Institute (RPMI) medium. HeLa and HeLa-HBZ clonal cell lines [48], CHO-LFA-1 clones [38], and HEK293T cells were cultured in Dulbecco’s modified Eagle’s medium (DMEM). All cells were supplemented with 10% FBS or 10% FetalPlex (GeminiBio) and 2 mM L-glutamine, 100 U/mL penicillin, and 50 μg/mL streptomycin. Jurkat pminLuc-viral CRE cells [38] and Jurkat-HBZ cells [38] were supplemented with 1.5 mg/mL of G418. HeLa and CHO clones were supplemented with 0.5 mg/mL of G418. Primary lymphocytes and lymphocyte cell lines were cultured with IL-2. Primary lymphocytes were activated in culture wells coated with anti-CD3 and anti-CD28 antibodies. Where indicated, cells were treated with 10 μM A485 (MedChem Express) or DMSO for 3 h.

### 2.3. RNA Extraction, cDNA Synthesis, and Quantitative Real-Time PCR 

RNA was isolated from cells using TRIzol Reagent (Invitrogen, Waltham, MA, USA) and cDNA was synthesized with random hexamers using the iScript cDNA Synthesis Kit (Bio-Rad, Hercules, CA, USA) or the Revert Aid kit (ThermoFisher Scientific, Waltham, MA, USA) as described by the manufacturers. Primer sequences are as follows: UBE2D2, 5′-TGCCTGAGATTGCTCGGATCTACA-3′ and 5′-ACTTCTGAGTCCATTCCCGAGCTA-3′; COL4A1, 5′-TCTGGCTGTGGCAAATGT-3′ and 5′-GGTAGTCCTGGTTCTCCAGTAT-3′; COL4A2, 5′-GCTTCTGGAAGGGCCAAT-3′ and 5′-CACGGCACATCAAACTTCTTC-3′; GEM, 5′-AATGAATGGCTCCATGACCACTGC-3′ and 5′-CTTGCAGTCAAACACCACTGCACA-3′; and NRP1c 5′-CAGAGCGCTCCCGCCTGAAC-3′ and 5′-AAATGGCGCCCTGTGTCCCG-3′. Real-time PCR was performed using iTaq Universal Supermix (Bio-Rad, Hercules, CA, USA) and a CFX Connect Real-Time PCR Detection System (Bio-Rad, Hercules, CA, USA), and relative mRNA levels were determined as described [48]. Serial dilutions of an appropriate experimental sample were used to generate standard curves for all primer sets included on a PCR plate. From the compilation of all the standard curves for all primers and all PCR plates (analyses), including ChIP PCR plates, the amplification efficiencies ranged from 63.1 to 129% with correlation coefficients ranging from 0.935 to 0.999.

### 2.4. Western Blot Analysis 

Cells were normalized to 5 × 10^5^ cells/mL, cultured overnight, and harvested. Whole-cell extracts were prepared, and Western blotting was conducted as described [49]. Antibodies used were as follows: anti-His (ab9108), anti-Nrp1 (ab81321), and anti-MafG (ab154318) were purchased from Abcam; anti-β-actin clone C4 (MAB1501) was purchased EMD Millipore; anti-CBP (sc-1211), anti-p300 (sc 57865), anti-gp46 (sc 57865), and anti-p19 (sc 57870) were purchased from Santa Cruz; and anti-Tax (hybridoma 168B17-46-92) was obtained from NIH AIDS Research and Reagent Program. Blots were developed using Pierce ECL Plus (ThermoFisher Scientific) and scanned with a Typhoon RGB imager (Cytiva). Images were analyzed using ImageQuant TL v8.1 (GE Healthcare Lifesciences).

### 2.5. Flow Cytometry 

A total of 10^6^ cells/labeling reaction was collected via centrifugation at 800× *g* for 3 min at 4 °C, washed once in 2 mL of cold PBS/0.2% BSA (FACS buffer), and suspended in 50 μL of cold FACS buffer, to which 1 μg of anti-Nrp1 Alexa Fluor 647 (R&D Systems, FAB3870R) was added. Cells were labeled on ice for 1 h and then washed with 2 mL of FACS buffer. Cells were fixed with PBS/2% paraformaldehyde at 4 °C for at least 30 m, suspended in 500 μL FACS buffer, and analyzed using a Cytek Aurora flow cytometer (Cytek Biosciences). Data were analyzed using FlowLogic Software version 8.4. 

### 2.6. Chromatin Immunoprecipitation (ChIP) Assays

ChIP assays were performed using the ZymoSpin ChIP Kit (Zymo Research) according to the manufacturer’s instructions with minor modifications. For p300 and CBP immunoprecipitations, chromatin was crosslinked using 10 mM disuccinimidyl glutarate (ThermoFisher Scientific) for 45 m and then crosslinked with formaldehyde; for all other immunoprecipitations, only formaldehyde was used. Crosslinked chromatin was sonicated using a Misonix Sonicator 4000 (20 s pulse on, 30 s pulse off, amplitude 40, 5 m processing time). Each immunoprecipitation reaction contained 5 μg of antibody and 200 μg of crosslinked sonicated chromatin. Antibodies used were as follows: anti-p300 (C-20, sc-585) from Santa Cruz Biotechnology; anti-CBP (D6C5, #7389), anti-JunB (C37F9, #3753), and anti-c-Jun (60A8, #9165) from Cell Signaling Technology; anti-MafG (ab154318) from Abcam. HBZ was immunoprecipitated through its C-terminal 6xHis tag using an anti-6xHis antibody (Abcam, ab9108). Purified ChIP DNA was amplified in iTaq Universal Supermix (Bio-Rad) using a CFX Connect Real-Time PCR Detection System (Bio-Rad). Primer sequences are as follows: NRP1 HBZ peak 5′-GCCAGTTCAGTACCCAGTAATA-3′ and 5′-CTGGAAATTAAGGTGGCTGTTT-3′; NRP1 off-target 5′-CTGAGACTTCTGGAGGCTAAAT-3′ and 5′-GGTATCCCAAATTCCCAGAGT-3′; WEE1AP1 5′-CCAATCGGCTTATCGGCTTAT-3′ and 5′-ACAGGAGCGTGTTTAGGTATTG-3′. Standard curves were generated for primer sets using 10-fold serial dilutions of each input DNA from the ChIP procedure and were included on each experimental plate. Enrichment values were quantified relative to the input as described [50,51].

### 2.7. Small RNA Interference 

The siGENOME SMART pool M-003486-04-0005 and M-003477-02-0005 were used to knock down p300 and CBP, respectively, while the siGENOME Non-Targeting siRNA pool#1 D-001206-13-05 was used as a control (Dharmacon). Cells were seeded to reach ~50% confluence on the day of transfection. Cells were transfected with 25 nM of siRNA using DharmaFECT 1 siRNA transfection reagent (Dharmacon) according to the manufacturer’s instructions. The medium was changed 24 h after transfection, and cells were cultured for an additional 48 h prior to harvesting.

### 2.8. Transfection and Single-Cycle, Replication-Dependent Infection Assays 

Single-cycle, replication-dependent luciferase assays were performed using Jurkat-HBZ or HEK293T cells as effector cells. Jurkat-HBZ cells (3 × 10^6^) were electroporated with 4.5 μg of pCMVHT1 [45] or pcDNA3.1 and 8 μg of pCRU5HT1-inLuc [45], 1.25 μg of pSG-Tax [43], and 1.25 μg of shRNA expression vector in 300 μL of RPMI/10 mM dextrose/0.1 mM dithiothreitol per 0.4 cm electroporation cuvette. Cells were exposed to a single exponential decay pulse of 200 V/950 µF. Forty-eight hours after electroporation, 5 × 10^5^ transfected Jurkat-HBZ cells were co-cultured with 8 × 10^4^ CHO-LFA-1 cells for 3 h. Jurkat-HBZ cells were then removed, and the CHO-LFA-1 cells were washed four times with PBS. CHO-LFA-1 cells were cultured for an additional 48 h, washed with PBS, and lysed with Passive Lysis Buffer (Promega, Madison, WI, USA). HEK293T cells were plated at 2.4 × 10^5^ cells/well in 24-well plates the day before transfection. Cells were transfected with 1.12 μg of pCMVHT1, 1.68 μg of pCRU5HT1-inLuc, and 1.2 μg of pQCXIP, pQCXIP-NRP1 or pQCXIP-NRP1-Δabc using TurboFect (ThermoFisher Scientific, Waltham, MA, USA) as described by the manufacturer. The cells were washed with PBS and lysed with Passive Lysis Buffer (Promega) 48 h later. Luciferase activity was measured using the luciferase assay system (Promega) and a GloMax 20/20 luminometer (Promega). Luminescence values were normalized to protein concentrations. HeLa cells were plated at 2.4 × 10^4^ cells/well in a 6-well plate and cultured overnight. Cells were then transfected with 4 μg of pSG5 or pSG-Tax-His using TurboFect (ThermoFisher Scientific, Waltham, MA, USA) as described by the manufacturer.

### 2.9. Lentiviral Transduction 

Lentivirus transductions were performed as described [37] but with the following modification: 51 µg of shRNA or pLJM1 expression vectors were used; media of transfected HEK293T cells were replaced with 10 mL IMDM supplement with 5% FBS to concentrate virus from viral supernatants using LentiX Concentrator (Takara Bio USA, San Jose, CA, USA). Cells were placed under puromycin selection (MT-2, 2 μg/mL; SLB-1, 6 μg/mL; ATL-2, 0.5 μg/mL) three days later. Cells were processed for co-culture/infections, Western blotting, and/or ELISA following three to four days of antibiotic selection. Co-culture/infection assays were conducted as described [37].

### 2.10. Isolation of Virions and Detection of Gag p19 in Culture Media 

HTLV-1-infected T-cells, ATL-2, MT-2 and SLB-1, were cultured at 1 × 10^6^ cells/mL overnight. Supernatants were collected via centrifugation at 1300 RPM for 3 min at room temperature and filtered through a 0.2 µm PES filter to ensure the complete removal of cells. Supernatants were centrifugated in an SW-40 Ti rotor (Beckman Coulter Life Sciences, Brea, CA, USA) at 20,000 RPM for 2 h at 4 °C. Concentrated virus was collected in 2x sodium dodecyl sulfate dye for Western blot analysis. For ELISA detection, cells were equalized and cultured for 24–48 h. Supernatants were collected and passed through 0.45 µm PES filters, and the virus was inactivated at 55 °C for 30 min. An HTLV p19 Antigen ELISA (ZeptoMetrix, Buffalo, NY, USA) kit was used as described by the manufacturer. Absorbances were detected with an accuSkan FC (Fisher Scientific, Hampton, VA, USA). 

### 2.11. In Silico Analysis and Statistical Analysis

Microarray data sets used in this study are available at NCBI Gene Expression Omnibus (GEO): GSE94409 [52]. For each sample, probes corresponding to the *COL4A1*, *COL4A2*, *GEM*, and *NRP1* transcripts in KK1 and ST1 cells infected with Ctrl, HBZ_1, or HBZ_2 sgRNAs were identified, and GEO2R was used to obtain expression values. ChIP-Seq data sets from GEO accession number GSE94732 [52] were analyzed using the Human Mar. 2006 (NCBI36/hg18) assembly with the IGV Browser [53]. Two-tailed Student’s t-tests were used for two-group comparisons, and significance was established at *p* < 0.05.

## 3. Results

### 3.1. Genes Involved in HTLV-1 Infection Are Upregulated in HBZ-Expressing Cells

HBZ was shown to enhance HTLV-1 infection by activating the transcription of *ICAM1* and *MYOF* [37,38]. Analysis of previous gene expression microarray data using HeLa cell clones lacking or expressing HBZ [48] revealed three additional genes potentially upregulated by HBZ that were previously reported to be involved in viral infection. These included *COL4A1*, *GEM*, and *NRP1*. *COL4A1* along with *COL4A2* encode collagen type IV alpha 1 and 2 chains, respectively, and are expressed in HTLV-1-infected T-cells [40]. These specific collagen proteins likely form the collagen matrix of biofilm-like viral assemblies on HTLV-1-infected cells that transfer virions to target cells during cell contact [8]. Gem is a GTP-binding protein with roles in signal transduction [54]. In HTLV-1-infected T-cells, Gem promotes the formation of cell-to-cell conjugates, potentially increasing infection of target T-cells [41]. Lastly, NRP1 encodes neuropilin-1 (Nrp1), which is the cellular receptor that interacts with the HTLV-1 envelope protein to facilitate the stable binding of virions to target cells [55]. Using quantitative reverse transcriptase PCR (qRT-PCR), expression of all three genes was confirmed to be elevated in the HBZ-expressing cells (Figure 1A,C,D). Given that *COL4A1* and *COL4A2* are “head-to-head” genes, they are expected to share certain promoter elements, which led us to also analyze *COL4A2* expression. The expression of this gene was also slightly elevated in HBZ-expressing cells (Figure 1B). It is important to note that *COL4A1/A2* expression is high in HeLa cells, which might partially mask levels of activation by HBZ. Regulation of *COL4A1* and *GEM* expression by HBZ was partly supported by in silico analysis of RNA-seq data from Nakagawa et al., who used CRISPR-Cas9 to disrupt the *hbz* gene in ATL cells (Appendix A; [52]). Furthermore, analysis of ChIP-seq data from the same study revealed peaks of HBZ-enrichment associated with each gene (Appendix A; [52]).

### 3.2. Nrp1 Expression Is Elevated in HTLV-1-Infected T-Cells Lines and Primary Cells Infected with HTLV-1

Roles for COL4A1, COL4A2, and Gem in HTLV-1 infection have been investigated. In contrast, in the context of expression by the infected cell, whether Nrp1 participates in viral infection is not known. This point led us to pursue further analyses of Nrp1. We first correlated higher mRNA levels with higher protein levels in HeLa cells expressing HBZ compared to cells carrying the empty vector (Figure 1E). In addition to HBZ, the HTLV-1 Tax protein is a transcriptional regulator [56]. However, unlike HBZ, the expression of Tax in HeLa cells did not lead to an increase in the level of Nrp1 (Appendix A). In silico analysis of the microarray data from the HBZ knockout cells of the Nakagawa et al. study [52] revealed reduced *NRP1* mRNA levels by each of the two *hbz*-targeted guide RNAs used in the two ATL cell lines tested (Figure 2A). Western blot analysis confirmed that Nrp1 is present in HTLV-1-infected T-cell lines, with the highest levels of the protein found in ATL-2 and MT-2 cells (Figure 2B). Importantly, flow cytometric analysis confirmed the presence of Nrp1 on the surface of infected cells (Figure 2C). qRT-PCR analysis also indicated that Nrp1 is expressed in recently established HTLV-1-immortalized cell lines from human peripheral blood lymphocytes ([47]; Figure 2D). Interestingly, some cell lines exhibited substantially higher levels of *NRP1* mRNA than ATL-2 cells. This comparison of recently established HTLV-1-immortalized cells and long-term HTLV-1 cell lines suggests that Nrp1 can be activated early in the course of infection.

### 3.3. HBZ Activates NRP1 Transcription from an Enhancer Downstream of the Gene

To characterize the mechanism through which HBZ upregulates *NRP1* transcription, we first analyzed ChIP-seq data from the Nakagawa et al. study [52]. In the two cell lines examined in this study, a peak of HBZ enrichment was identified approximately 50 kb downstream of *NRP1* (Figure 3A). This site is also 176 kb upstream of *ITGB1* (Appendix A); however, the expression of this gene is not affected by HBZ according to microarray data. To test for HBZ enrichment at this site in another HTLV-1-infected T-cell line, we performed ChIP assays using SLB-1 cells. As antibodies against HBZ that are suitable for ChIP assays have not been developed, cells were transduced to express HBZ with a C-terminal 6xHis epitope tag for immunoprecipitation. From this approach, we observed significant enrichment of HBZ at the peak region identified in KK1 cells compared to a downstream off-target region (Figure 3B). These results support our analysis of data from the previous study, showing that HBZ is recruited to a chromosomal region downstream of the *NRP1* gene. A general analysis of this region using the UCSC genome browser revealed that it is denoted as an enhancer independent of HBZ and HTLV-1 infection [57,58]. Specifically, it shows hypersensitivity to DNase I and is flanked by peaks of histone H3 lysine 27 (H3K27ac; Appendix A). Given that we did not confirm this region as a *bona fide* enhancer in an enhancer-based-reporter assay, it is necessary to deem this region as a candidate enhancer (herein referred to as an enhancer).

Recent evidence indicates that, via dimerization with small Mafs and Jun proteins, HBZ can associate with the DNA [32,33,37]. We, therefore, analyzed the enrichment of these proteins at the downstream enhancer. ChIP analysis of ATL-2 cells revealed significant enrichment of c-Jun and JunB at this site compared to the off-target region (Figure 3C). Strikingly, the level of enrichment of these proteins at the enhancer matched that at the AP-1 site in the *WEE1* promoter, which served as the positive control. Consistent with these results, the DNA sequence of the enhancer region contained two full consensus AP-1 binding sites as well as several partial sites (Appendix A). No significant enrichment was detected for the small Maf, MafG, at the enhancer (Figure 3C).

We then used HeLa clonal cell lines to expand on these observations. First, we compared *NRP1* mRNA levels in a clone expressing wild-type HBZ and two clones expressing mutant versions of the viral protein: HBZ-MutAD, which is defective for binding to p300/CBP, and HBZ-MutZIP, which is defective for binding to cellular bZIP factors [24,32]. In addition, we analyzed a start codon mutant (HBZΔATG) that is not translated into the viral protein [29]. Cell lines expressing either HBZ-MutZIP or HBZΔATG showed a significant reduction in *NRP1* mRNA levels compared to cells expressing wild-type HBZ, while no significant change was observed with HBZ-MutAD (Figure 3D). Using ChIP assays, we verified that HBZ was enriched at the *NRP1* enhancer in the HeLa cells expressing wild-type HBZ (Figure 3E). Lastly, using ChIP assays to compare the HeLa clones expressing HBZ and carrying the empty expression vector, we observed the enrichment of JunB at the enhancer in the presence of HBZ (Figure 3F). Together, these results indicate that HBZ activates *NRP1* transcription by forming heterodimers with Jun proteins on the enhancer.

Analysis of ChIP-seq data from the Nakagawa et al. study [52] also revealed peaks of H3K27ac at and around the *NRP1* enhancer (Figure 3A). This modification is generated via the KAT activity of p300 and CBP, suggesting the involvement of these coactivators in HBZ-mediated *NRP1* transcription. In the HeLa clones, ChIP assay results revealed that both p300 and CBP were enriched at the enhancer compared to at the off-target region with substantially greater coactivator enrichment in the presence of HBZ (Figure 4A,B). Consistent with this observation, siRNA-mediated knockdown of both coactivators reduced *NRP1* mRNA levels in both the HBZ-expressing and empty vector clones (Figure 4C). Knockdown of p300 and CBP was confirmed via Western blot (Figure 4D). In ATL-2 cells, treatment with the p300/CBP KAT-specific inhibitor, A485, significantly reduced *NRP1* mRNA levels (Figure 4E). A similar effect of A485 was observed in a recently established HTLV-1-immortalized cell line (Figure 4F). Together, these results indicate that HBZ upregulation of *NRP1* transcription is associated with the enhanced recruitment of p300/CBP to the downstream enhancer.

### 3.4. Nrp1 Expression in HTLV-1-Infected T-Cells Inhibits HTLV-1 Infection

We were interested in establishing whether Nrp1 expressed by HTLV-1-infected T-cells influenced HTLV-1 infection. To test this possibility, we first analyze how knocking down Nrp1 expression in these cells influenced their ability to infect target reporter cells. In these experiments, we used MT-2 and ATL-2 cells based on their higher Nrp1 expression compared to SLB-1 cells. These two effector cell lines were transduced to express shRNA targeting the *NRP1* transcript (shNRP1) or, as a negative control, GFP (shGFP) and then co-cultured with Jurkat-pminLUC-vCRE reporter cells (Figure 5A). Following HTLV-1 infection, cells express the viral protein, Tax, which trans-activates the promoter driving luciferase expression in the Jurkat-pminLUC-vCRE cells [38]. C8166/45 cells were used as negative control effector cells, as they do not produce HTLV-1 virus particles due to defects in their proviruses [59]. In MT-2 cells, the knockdown of Nrp1 led to a significant increase in HTLV-1 infection over that of shGFP-transduced cells (Figure 5B). Western blot analysis showed that levels of the HTLV-1 structural protein, Gag p19, and the HTLV-1 envelope surface unit (SU), gp46, were not affected by Nrp1 knockdown (Figure 5C). Furthermore, clarified culture supernatants contained similar levels of Gag p19, indicating that Nrp1 knockdown does not affect the production of HTLV-1 virus particles (Figure 5D). Comparable results were obtained using transduced ATL-2 cells (Figure 5E–G). 

We additionally analyzed Jurkat cells stably expressing HBZ (Jurkat-HBZ) as effector cells using a single-cycle, replication-dependent luciferase infection assay [45]. We used these cells based on the strong luciferase signal they generate in target cells [38]. For these assays, Jurkat-HBZ cells were co-transfected with an HTLV-1 packaging vector (pCMVHT1M), the replication-dependent HTLV-1 reporter vector (pCRU5HT1-inLuc), and to increase infection efficiency, a Tax expression vector. To examine the effects of Nrp1 in this system, cells were additionally co-transfected with the shRNA expression vectors targeting *NRP1* or GFP transcripts. Transfected cells were co-cultured with adherent CHO-LFA-1 target cells and then removed, and luciferase activity was measured in the CHO-LFA-1 cells (Figure 5H). As target cells, CHO-LFA-1 express lymphocyte function-associated antigen 1 (LFA-1), which binds ICAM-1 on effector cells to stabilize cell-to-cell contact and induces the formation of a virological synapse from which infection occurs [9]. With this approach, we observed that Jurkat-HBZ cells co-transfected with the shRNA vector targeting *NRP1* produced a higher level of infection than cells co-transfected with the shGFP control vector (Figure 5I).

We performed reciprocal experiments using SLB-1 cells, which display low Nrp1 expression compared to ATL-2 and MT-2 cells. Cells were transduced with an Nrp1 expression vector or the empty expression vector and then co-cultured with Jurkat-pminLUC-vCRE reporter cells (Figure 6A). Western blot results confirmed higher Nrp1 expression in the cells transduced with the Nrp1 expression vector compared to those transduced with the empty vector (Figure 6B). Consistent with the knockdown experiments, Gag p19 and gp46 were not affected by the variations in the level of Nrp1. However, higher Nrp1 expression was associated with a significant decrease in HTLV-1 infection (Figure 6C). These and the Nrp1 knockdown results indicate that Nrp1 expressed by effector cells has an inhibitory role in HTLV-1 infection.

### 3.5. Nrp1 Is Incorporated into the Viral Particle

Consistent with our findings, a recent study demonstrated that, when expressed in effector cells, Nrp1 inhibits HIV infection [60]. This effect was found to be due to the incorporation of Nrp1 into HIV virions, which led us to test whether Nrp1 is similarly incorporated into HTLV-1 virions. Western blot analysis revealed the presence of Nrp1 in cell-free HTLV-1 virions isolated via ultracentrifugation (Figure 7A). Given the extended extracellular structure of Nrp1 and its heavily glycosylated state, the authors of the previous study proposed that Nrp1 may sterically disrupt the binding of HIV virions to target cells. To address this hypothesis in the context of HTLV-1, we analyzed an Nrp1 deletion mutant lacking most of its extracellular region (Figure 7B,E). HEK293T cells were transfected with the set of single-cycle, replication-dependent luciferase infection assay plasmids and co-transfected with an expression vector for full-length Nrp1 or the deletion mutant (Figure 7C). Subsequent analysis of luciferase activity from the cultures revealed a significant decrease in infection in cultures with ectopic expression of full-length Nrp1, while cultures with the deletion mutant showed no change in infection (Figure 7D). These results show that the extracellular region of Nrp1 is important for impairing infection and may be due to the occlusion of virion–target cell interactions during infection through cell-to-cell contact. 

## 4. Discussion

HBZ was previously shown to enhance HTLV-1 infection by activating the expression of *ICAM1* and *MYOF* [37,38], and in this study, we found that HBZ upregulates two additional cellular genes involved in infection: *COL4A1* and *GEM* [40,41]. Interestingly, apart from *MYOF*, these genes are also activated by the HTLV-1-encoded protein, Tax, which plays an essential role in HTLV-1 infection [9]. While the interplay between Tax and HBZ in infection has not been addressed, it is possible that both proteins act together to augment the expression of these genes. Alternatively, HBZ may play a supporting role to maintain some level of HTLV-1 infectivity when Tax expression switches to the off state. Indeed, in an HTLV-1-induced leukemic cell line, Tax expression was found to stochastically alternate between on and off states [61]. Moreover, when Tax is in the off state, virus particles may be retained on the surface of the cell in an extracellular matrix [8] and poised for infection. Finally, while mitotic expansion appears to be the primary mode of viral replication once the adaptive immune response is activated and a proviral set point is established, some infectious spread persists [62,63,64]. 

In addition to these genes, HBZ upregulated the expression of Nrp1, which on target cells, serves as the high-affinity binding receptor for HTLV-1 virions [13]. While the significance of *NRP1* expression by HTLV-1-infected T-cells has not been reported, *NRP1* expression was found to be upregulated in mouse primary CD4^+^ T-cells transduced to express HBZ [65]. This observation prompted us to explore this gene further. We first analyzed how HBZ upregulates *NRP1* transcription. A peak of HBZ enrichment was identified approximately 50 kb downstream of the gene. Interestingly, independent of HBZ or HTLV-1 infection, cumulative data from multiple cell specimens show that this chromosomal region acts as an enhancer [58]. For example, it comprises a DNase I hypersensitive peak tightly flanked by peaks of histone H3 lysine 27 acetylation. These features are indicative of a nucleosome-free region bound by transcriptional regulators, including p300/CBP, that acetylates H3K27. The presence of p300 has been found to be a common feature of enhancers [66,67].

We speculate that HBZ primarily serves to increase the association of Jun members (i.e., c-Jun, JunB, and JunD) with the enhancer. HBZ is known to form heterodimers with these factors through interactions between the leucine zipper (ZIP) domain of each protein [27,28,33], and mutations in the ZIP domain of HBZ that disrupt Jun protein-binding also abrogated *NRP1* transcription. The enhancer contains two consensus AP-1 binding sites as well as multiple partial sites. While heterodimers formed between HBZ and a Jun member may bind one or both consensus AP-1 sites, it is alternatively possible that such heterodimers target an AP-1 partial site. AP-1 transcription factors bind DNA through the basic region of the bZIP domain of each subunit [68], and in HBZ, this region lacks the conserved amino acid motifs involved in binding the AP-1 sequence. Therefore, in the context of an HBZ/Jun member heterodimer, the Jun member may contact an AP-1 half site while HBZ contacts an adjacent unrelated sequence. The observation that JunB is enriched at the enhancer in the absence of HBZ suggests that the cellular AP-1 factors are binding the consensus AP-1 sites.

It is possible that a second mechanism also contributes to the increased association of Jun members with the enhancer that involves an increased abundance of these proteins in the presence of HBZ. We reported that the splice 1 variant of HBZ (HBZ_S1_), which was used in this study, stabilizes c-Jun and JunB by inhibiting their proteosomal degradation initiated by the E3 ubiquitin ligase, constitutive photomorphogenesis protein 1 [69]. Of note, HBZ_S1_ is the most abundant variant in HTLV-1-infected T-cells [70,71,72,73]. 

In contrast to mutations in the ZIP domain, mutations in the AD of HBZ did not significantly affect transcription. This observation diverges from some previous results in which the AD has been shown to be central to transcriptional activation by HBZ [37,48,74,75]. While the AD appeared to be dispensable for activating *NRP1* transcription, both p300 and CBP displayed higher levels of association with the enhancer in the presence of HBZ. This observation might suggest that the increased association of Jun members with the enhancer augments recruitment of p300/CBP.

In addition to this proposed model, there are likely HBZ-independent mechanisms contributing to the regulation of *NRP1* transcription in HTLV-1-infected T-cells. Indeed, there was a wide range of *NRP1* expression levels among the HTLV-1 T-cells lines and HTLV-1-immortalized clones we tested that did not necessarily reflect HBZ expression levels according to our previous results [37,48,76]. Of note, SLB-1 cells did not express Nrp1; however, our data indicate that HBZ is recruited to the enhancer in these cells. These observations suggest that another critical component of Nrp1 expression is altered or missing in these cells. Overall, we cannot explain the variation in transcript and protein levels but suspect it may relate to genetic heterogeneity or epigenetic differences across the cell lines and clones.

In this study, we approached Nrp1 expression in HTLV-1-infected T-cells based on its role as the viral receptor that forms a high-affinity interaction with SU. The ability of HBZ to increase the expression of one of the HTLV-1 receptors appears to oppose conventional replication strategies used by some avian retroviruses and HIV. Expression of the viral receptor on cells infected with these retroviruses promotes reinfection, leading to the accumulation of unintegrated DNA, which is cytopathic [77,78]. Through multiple virus-mediated mechanisms, HIV has been shown to generally prevent reinfection by eliminating CD4 from the surface of the infected cell [79]. While fusion and virus entry for HIV requires the chemokine receptors CCR5 or CXCR4 [80], for HTLV-1, these processes are believed to require Glut1 [13]. Interestingly, the HTLV-1 protein, Tax, was shown to reduce Glut1 at the cell surface by binding sorting nexin 27 (SNX27) and preventing SNX27 from trafficking Glut1 to the cell surface [81]. Therefore, HTLV-1 reinfection might be impaired by the removal of Glut1 rather than Nrp1 from the plasma membrane.

We found that the expression of Nrp1 on HTLV-1-infected T-cells and HTLV-1-producing cells was associated with decreased viral infection without any significant effect on viral production. A similar observation was reported recently regarding HIV-infected cells of the monocyte lineage [60]. In this other study, Nrp-1 expressed by macrophages and dendritic cells was found to be packaged into the HIV virions produced by these cells, leading to reduced binding of these virions to target cells. The authors of this study speculated that the extended ectodomain of Nrp1 along with its heavily glycosylated state sterically inhibits the attachment of virions to target cells. We similarly found that Nrp1 is incorporated into HTLV-1 virions, and consistent with the hypothesis of steric inhibition, an Nrp1 mutant lacking most of the ectodomain did not reduce viral infection. Therefore, Nrp1 might also reduce the binding of HTLV-1 virions to target cells. Limitations of our experimental models prevented the comparison of virions produced by cells expressing Nrp1 versus those lacking Nrp1. Specifically, following the transduction of HTLV-1-infected T-cells, we were unable to obtain a sufficient quantity of viable cells to isolate cell-free virions. 

The negative effect of Nrp1 on HTLV-1 infection appears to be outweighed by positive contributions from other HBZ-regulated genes, at least in the cell culture models we have tested [37,38]. It is possible that in some HTLV-1 carriers, host genetic factors participate with HBZ to increase Nrp1 expression. Indeed, we found variability in *NRP1* transcript levels in the different HTLV-1-immortalized cell lines (from different blood donors) and, via analysis of GEO datasets, in CD4^+^ T-cells/PBMC from different HTLV-1 carriers (data not shown). Perhaps, Nrp1 might impact infection in cases where it is more highly expressed. Current evidence indicates that the infectious spread of the virus is associated with the risk of disease development [82,83]. Therefore, Nrp1 expression levels in CD4^+^ T-cells of HTLV-1 carriers might serve as a biomarker to gauge the risk of disease development.

Finally, it is possible that Nrp1 contributes to other aspects of HTLV-1 biology not addressed in this study. One example involves the role of Nrp1 as a coreceptor for TGF-β receptor signaling [20,21], which is interesting considering that HBZ activates transcription through the downstream signaling effector, Smad3 [75]. In addition, Nrp1 is capable of converting the latent form of TGF-β into the active form [20]. Therefore, increasing Nrp1 expression might represent a second mechanism by which HBZ enhances TGF-β signaling. This signaling pathway is implicated in establishing the regulatory T-cell-like phenotype documented for most HTLV-1 infected cells [84]. It would be interesting to investigate this, and other potential effects of Nrp-1 related to HTLV-1 infection and pathogenesis. Indeed, small molecule inhibitors of Nrp1 that are in development as anticancer agents [85] may serve as a future therapeutic option for patients with HTLV-1.

## Figures and Tables

**Figure 1 pathogens-12-00831-f001:**
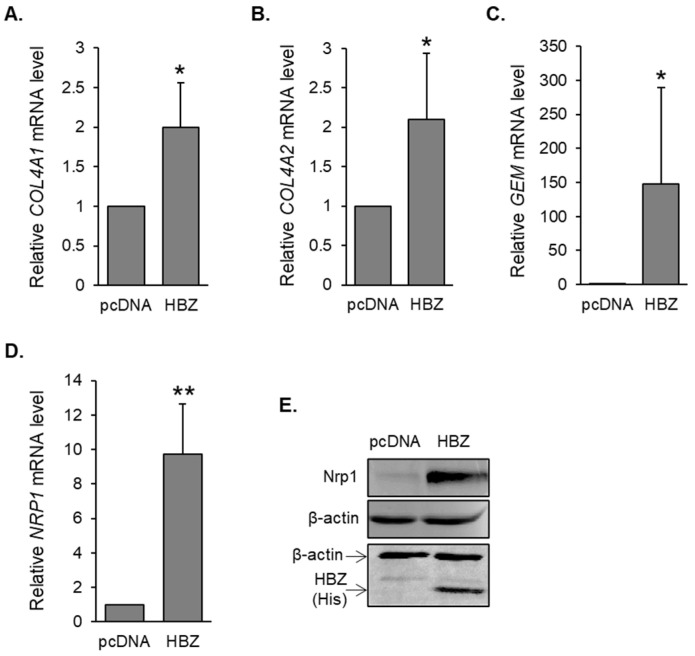
HBZ upregulates genes involved in HTLV-1 infection. (**A**) Relative *COL4A1* mRNA levels in HeLa clonal cell lines expressing wild-type HBZ (HBZ) or carrying the empty expression vector (pcDNA). The graph shows qRT-PCR results averaged from four independent experiments. (**B**) Relative *COL4A2* mRNA levels in HeLa clonal cell lines expressing wild-type HBZ (HBZ) or carrying the empty expression vector (pcDNA). The graph shows qRT-PCR results averaged from six independent experiments. (**C**) Relative *GEM* mRNA levels in HeLa clonal cell lines expressing wild-type HBZ (HBZ) or carrying an empty expression vector (pcDNA). The graph shows qRT-PCR results average from eight independent experiments. (**D**) Relative *NRP1* mRNA levels in HeLa clonal cell lines expressing wild-type HBZ (HBZ) or carrying an empty expression vector (pcDNA). The graph shows qRT-PCR results average from five independent experiments. For all graphs, HBZ values are normalized to that of the empty vector (set to 1), and error bars show standard deviations; * *p* < 0.05; ** *p* < 0.01. (**E**) Nrp1 expression in empty vector (pcDNA) and HBZ-HeLa clones. Whole-cell extracts (40 μg for Nrp1 and β-actin, 20 μg for His and β-actin) were analyzed via Western blot using antibodies against Nrp1, HBZ (6xHis tag), and β-actin.

**Figure 2 pathogens-12-00831-f002:**
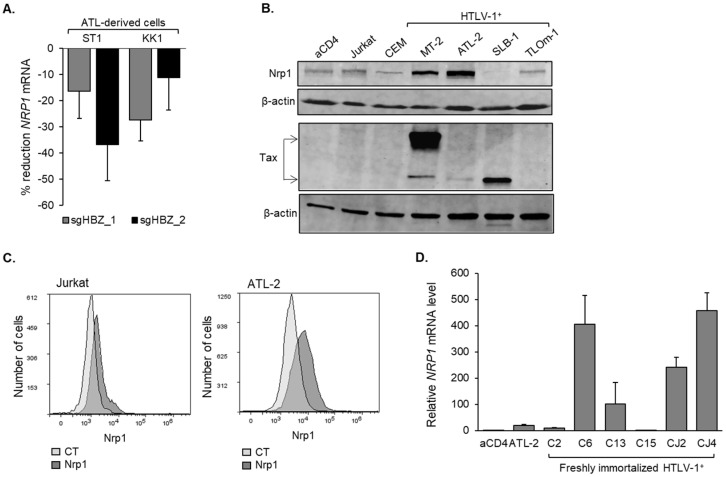
HBZ upregulates *NRP1* expression. (**A**) Deletion of HBZ in ST1 and KK1 ATL-derived cells reduces *NRP1* expression. The graph was generated from published microarray data (GEO accession number GSE94409 [52]) and shows the percent reduction in *NRP1* transcript levels after inducing CRISPR/Cas9-mediated knockout of HBZ in the ATL-derived cell lines, ST1 and KK1, using two different guide RNAs (sgHBZ_1 and _2). Values are from day 8 post-induction except for sgHBZ_2 in KK1, which is the day 7 value (no day 8 data were provided for this specimen). Data were obtained using GEO2R with calculations based on averaged values from the four array features probing for different regions of the *NRP1* transcript. (**B**) Nrp1 expression in non-infected activated CD4^+^ T-lymphocytes (aCD4) and T-cell lines. Whole-cell extracts (45 μg for Nrp1 and β-actin, 50 μg for Tax and β-actin) were analyzed via Western blot using antibodies against Nrp1, Tax, and β-actin. (**C**) Nrp1 expression on the cell surface of T-cell lines. Jurkat and ATL-2 cells were labeled with an Nrp1 antibody, fixed, and analyzed using flow cytometry. Histograms are representative of three independent experiments and show relative cell surface labeling as follows: unlabeled cells (CT, light grey) and Nrp1 antibody (dark gray). (**D**) Relative *NRP1* mRNA levels in HTLV-1-immortalized human T-cell lines recently established from peripheral blood lymphocytes (PBL). The graph shows qRT-PCR results averaged from three separate RNA extractions. Values were normalized to those for activated CD4^+^ T-cells (set to 1). Error bars represent standard deviations.

**Figure 3 pathogens-12-00831-f003:**
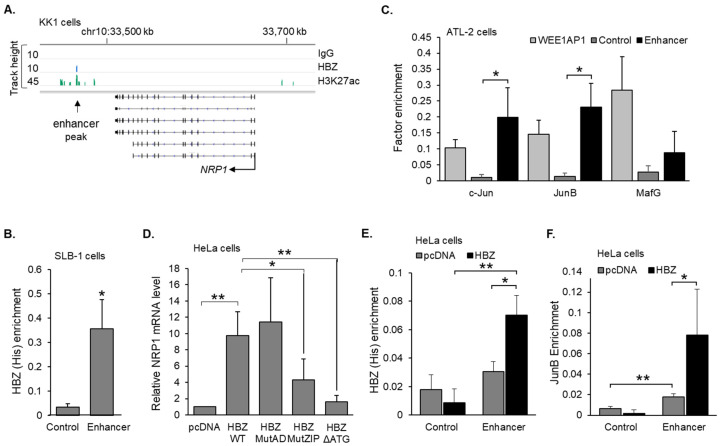
HBZ increases c-Jun and JunB recruitment to an enhancer downstream of the *NRP1* gene. (**A**) HBZ associates with a chromosomal site (enhancer peak) approximately 200kb downstream of the *NRP1* transcription start site (indicated by the bent arrow). Peaks of enrichment for HBZ, H3K27ac, and IgG (negative control) at the *NRP1* locus in KK1 cells are shown in the IGV Browser. Genomic coordinates are based on the NCBI36/hg18 assembly. Data were obtained from published ChIP-Seq data sets (GEO accession number GSE94732 [52]). (**B**) HBZ binds to the enhancer region in SLB-1 cells. The graph shows levels of HBZ enrichment at the off-target control site and the enhancer region averaged from four independent ChIP assays using SLB-1 cells transduced to express HBZ with a C-terminal 6xHis tag. (**C**) c-Jun and JunB are enriched at the enhancer region in ATL-2 cells. The graph shows average levels of factor enrichment at the off-target control site, the enhancer region, and the AP-1 site in the *WEE1* promoter (WEE1-AP1). Data are from four (c-Jun) and three (JunB and MafG) independent ChIP assays. (**D**) Relative *NRP1* mRNA levels in HeLa clonal cell lines expressing wild-type HBZ (HBZ-WT), the activation domain mutant (HBZ-MutAD), the leucine zipper domain mutant (HBZ-MutZIP), the translational-defective mutant (HBZ-ΔATG), or carrying the empty expression vector (pcDNA). The graph shows qRT-PCR results average from five independent experiments, with values normalized to that for pcDNA (set to 1). (**E**) HBZ binds to the enhancer region in HeLa cells. The graph shows levels of HBZ enrichment at the off-target control site and the enhancer region averaged from three independent ChIP assays using HeLa cells expressing HBZ or carrying the empty vector (pcDNA). (**F**) JunB binds to the enhancer region in HeLa cells. The graph shows levels of HBZ enrichment at the off-target control site and the enhancer region averaged from three independent ChIP assays using HeLa cells expressing HBZ or carrying the empty vector (pcDNA). For all graphs, error bars show standard deviations; * *p* < 0.05, ** *p* < 0.01.

**Figure 4 pathogens-12-00831-f004:**
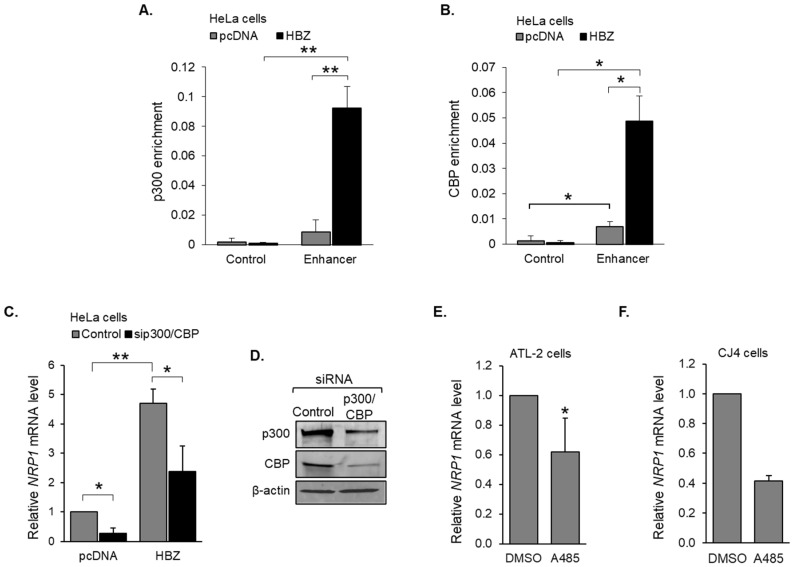
p300/CBP is recruited to the *NRP1* enhancer. p300 (**A**) and CBP (**B**) bind the *NRP1* enhancer region. Graphs show average values from three independent ChIP assays using empty vector (pcDNA) and HBZ-expressing HeLa cells. (**C**) siRNA-mediated depletion of p300 and CBP abrogates activation of *NRP1* transcription by HBZ. HeLa clonal cell lines expressing wild-type HBZ (HBZ) or carrying an empty expression vector (pcDNA) were transfected with an siRNA pool targeting p300 and CBP or a non-targeting siRNA pool (Control). The graph shows qRT-PCR results averaged from four independent transfection experiments with values normalized to those for the empty-vector clone (pcDNA) transfected with the non-targeting siRNA pool (set to 1). (**D**) siRNA-mediated depletion of p300 and CBP. HeLa cells were transfected with an siRNA pool targeting p300 and CBP (p300/CBP) or a non-targeting siRNA pool (Control). Whole-cell extracts (15 μg for p300, 40 μg for CBP and β-actin) were analyzed via Western blot using antibodies against p300, CBP, and β-actin. Inhibition of p300/CBP KAT activity reduces *NRP1* transcription in (**E**) an HTLV-1-infected T-cell line (ATL-2) and (**F**) an HTLV-1-immortalized primary human T-cell line (CJ4). Cells were treated with A485 (10 μM) or the carrier (DMSO) for 3 h. Graphs show qRT-PCR results averaged from four (ATL-2 cells) and two (CJ4 cells) independent experiments with A485 values normalized to those for DMSO (set to 1). For all graphs, error bars show standard deviations; * *p* < 0.05; ** *p* < 0.01.

**Figure 5 pathogens-12-00831-f005:**
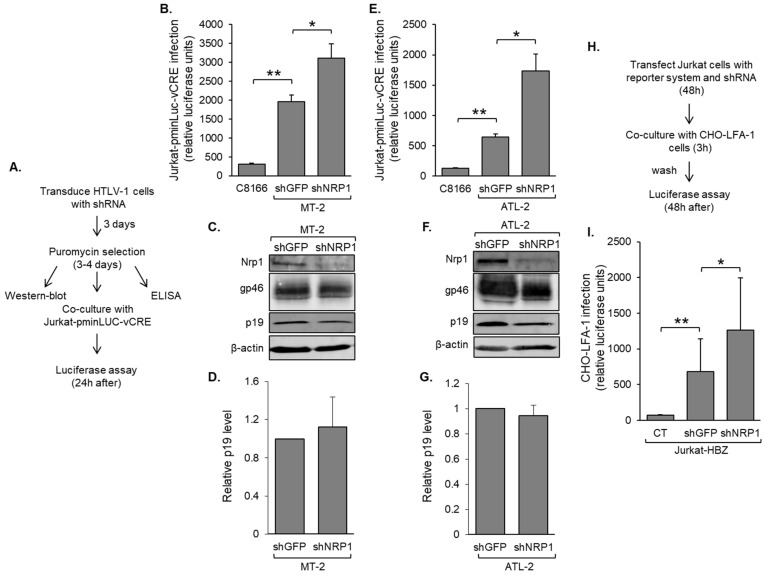
*NRP1* knockdown increases HTLV-1 infection. (**A**) The flow diagram shows the co-culture/infection assay procedure using HTLV-1-infected cells as donor cells and Jurkat-pminLUC-vCRE cells as target cells. (**B**) shRNA-mediated depletion of Nrp1 in MT-2 cells increases HTLV-1 infection. Jurkat-pminLUC-vCRE cells were co-cultured with MT-2 cells under puromycin selection following transduction with expression vectors for a negative control shRNA (shGFP) or an shRNA targeting the *NRP1* transcript (shNRP1), or co-cultured with non-infectious C8166/45 cells. The graph shows luciferase values averaged from three replicates of a single experiment and is representative of three independent experiments. (**C**) shRNA-mediated depletion of Nrp1 in MT-2 cells does not affect levels of gp46 (SU) and Gag p19. Whole-cell extracts (50 μg for Nrp1 and β-actin; 15 μg for gp46 and Gag p19) were analyzed via Western blot using antibodies against Nrp1, gp46, Gag p19, and β-actin. (**D**) shRNA-mediated depletion of Nrp1 in MT-2 cells does not affect levels of cell-free virus. Levels of Gag p19 in clarified culture media were measured using ELISA. The graph shows values averaged from two independent transduction experiments with shNRP1 values normalized to those for shGFP (set to 1). (**E**) shRNA-mediated depletion of Nrp1 in ATL-2 cells increases HTLV-1 infection. Experiments were performed as described in (**B**) above. The graph shows luciferase values averaged from three replicates of a single experiment and is representative of two independent experiments. (**F**) shRNA-mediated depletion of Nrp1 in ATL-2 cells does not affect levels of gp46 (SU) and Gag p19. Western blots were conducted as described in (**C**) above. (**G**) shRNA-mediated depletion of Nrp1 in ATL-2 cells does not affect levels of cell-free virus. Experiments were performed as described in (**D**) above. The graph shows values averaged from three independent transduction experiments with shNRP1 values normalized to those for shGFP (set to 1). (**H**) The flow diagram shows the co-culture/infection assay procedure using Jurkat cells as donor cells and CHO-LFA-1 cells as target cells. (**I**) shRNA-mediated depletion of Nrp1 in Jurkat donor cells increases HTLV-1 infection. Jurkat cells were co-transfected with pcDNA3.1, pCRU5HT1-inLuc, and pSG-Tax (no infection, CT) or pCMVHT1, pCRU5HT1-inLuc, pSG-Tax, and the shGFP or shNRP1 vector, and cocultured with CHO-LFA1 cells. The graph shows luciferase values normalized to protein averaged from replicates from three independent electroporation assays. For all graphs, error bars show standard deviations; * *p* < 0.05, ** *p* < 0.01.

**Figure 6 pathogens-12-00831-f006:**
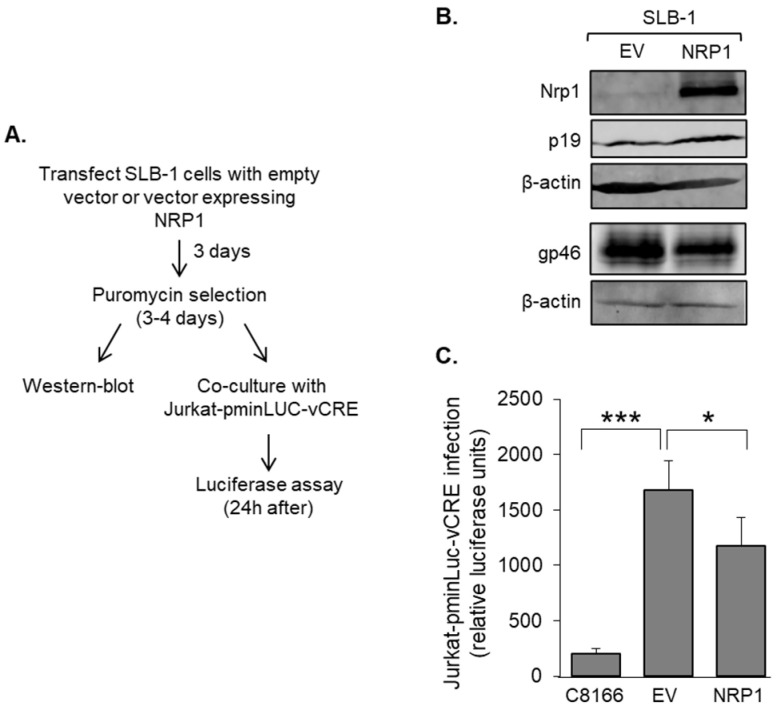
Overexpression of Nrp1 reduces infection. (**A**) The flow diagram shows the co-culture/infection assay procedure using HTLV-1-infected SLB-1 cells as donor cells and Jurkat-pminLUC-vCRE cells as target cells. SLB-1 cells were transduced with pLJM1-NRP1 (NRP1) or the pLJM1 empty vector (EV) and placed under puromycin selection. (**B**) Nrp1 expression in transduced SLB-1 cells. Whole-cell extracts (50 μg for Nrp1, Gag p19, and β-actin; 15 μg for gp46 and β-actin) were analyzed via Western blot using antibodies against Nrp1, gp46, Gag p19, and β-actin. (**C**) Increased expression of Nrp1 in SLB-1 cells decreases HTLV-1 infection. Jurkat-pminLUC-vCRE cells were co-cultured with SLB-1 cells transduced with pLJM1-NRP1 (NRP1) or the pLJM1 empty vector (EV), or co-cultured with non-infectious C8166/45 cells. The graph shows luciferase values averaged from three replicates of each infection condition from a single experiment and is representative of three independent experiments. Error bars show standard deviations; * *p* < 0.05, *** *p* < 0.001.

**Figure 7 pathogens-12-00831-f007:**
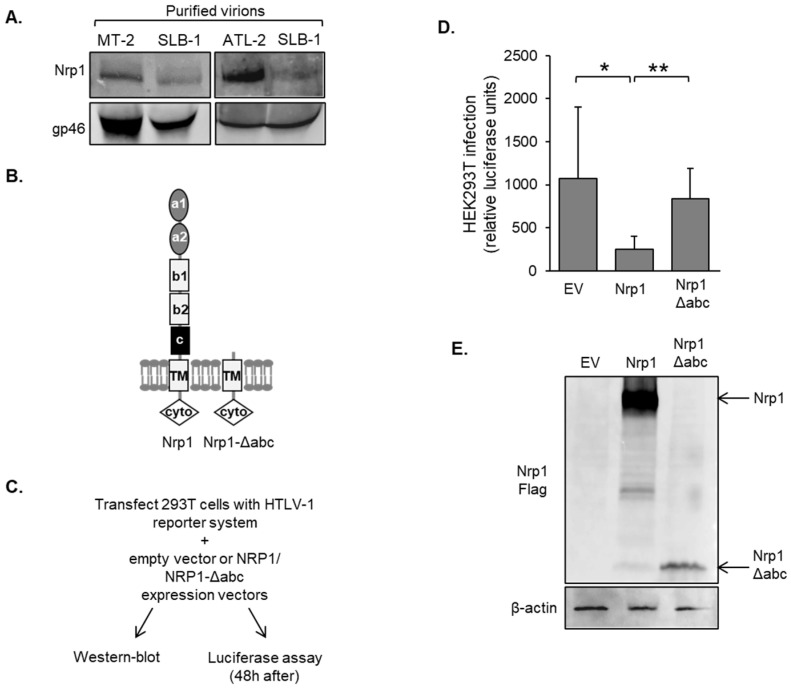
The ectodomain of Nrp1 is responsible for the inhibition of HTLV-1 infection. (**A**) Nrp1 is incorporated into HTLV-1 virus particles. Culture media from MT-2, SLB-1, and ATL-2 cells were filtered, ultracentrifuged, and analyzed via Western blot using antibodies against Nrp1 and gp46. (**B**) The schematic shows full-length Nrp1 and the truncation mutant, Nrp1-Δabc. (**C**) The flow diagram shows the co-culture/infection assay procedure using HEK293T cells. (**D**) HTLV-1 infection is not inhibited by an Nrp1 truncation mutant lacking the ectodomain. HEK293T cells were co-transfected with pCMVHT1M, pCRU5HT1-inLuc, and pQCXIP (EV), pQCXIP-NRP1, or pQCXIP-NRP1-Δabc. Luciferase assays were performed 48 h later. The graph shows luciferase values averaged from three independent experiments each performed in triplicate. Error bars show standard deviations; * *p* < 0.05; ** *p* < 0.01. (**E**) Nrp1 expression in transfected HEK293T cells. Whole-cell extracts (50 μg) were analyzed via Western blot using antibodies against Nrp1 (Flag-tagged) and β-actin.

## Data Availability

All data in this study are included in this article.

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
