# Peer review of "Upregulation of Neuropilin-1 Inhibits HTLV-1 Infection"

_pathogens, 2023, doi:10.3390/pathogens12060831_

Round 1

Reviewer 1 Report

Exceptional writing and details of this complex experiment to demonstrate the importance of Nrp1 and HBZ with HTLV-1.

Recommend adding additional information in the background with regard to HBZ's role/impact in the setting of HTLV-1 infection.  You pull in more information in the discussion, but think it may be helpful to introduce/elaborate earlier in the background to establish relevance.

Also, if possible, consider adding a graphic in the background section that visually displays the association of the major players regarding HTLV-1 infection.   It is such a complicated process that a visual would be helpful.  Perhaps this will already be done in a different HTLV-1 paper in this Special Issue and could be carried over to this paper?

Are there experimental limitations that should be detailed in this article?  For example, why were certain experiments repeated two times vs others that were repeated eight times.  Were the lesser repeated experiments that much more involved?  Were those repeated more frequently done to confirm more he experiment repeated 8 times (Figure 1, C) seemed to have a high standard deviation.  Could include other limitations if relevant.

Also, consider adding a conclusion and possible future directions for therapy options that this discovery could lead to?

Again, this was an extremely well written and organized, detailed in vivo experiment!!!

Author Response

We would like to thank all the reviewers for their positive feedback and constructive comments. To the best of our ability within this short revision period, we have attempted to answer their questions and address their comments, making some modifications to the manuscript. Our responses to the reviewers’ questions and comments (italicized) are below. Line numbers denoting where there are modifications to the manuscript correspond to the no mark-up version.

Recommend adding additional information in the background with regard to HBZ's role/impact in the setting of HTLV-1 infection.  You pull in more information in the discussion, but think it may be helpful to introduce/elaborate earlier in the background to establish relevance. In the Introduction, we added information regarding the critical role of Tax in the HTLV-1 process, suggesting that HBZ may augment the functions of Tax (line 77). We did not expand on this latter point more because we based the Introduction on published data and not new data regarding GEM and COL4A1.

Also, if possible, consider adding a graphic in the background section that visually displays the association of the major players regarding HTLV-1 infection.   It is such a complicated process that a visual would be helpful.  Perhaps this will already be done in a different HTLV-1 paper in this Special Issue and could be carried over to this paper? A graphic encompassing the three mechanisms of infection would be a challenge. We did not find another paper in this Special Issue covering this topic, so we referred to an excellent review of the infection process in a sister Journal (Gross and Thoma-Kress. 2016. Viruses, 8(3); line 45).

Are there experimental limitations that should be detailed in this article?  For example, why were certain experiments repeated two times vs others that were repeated eight times.  Were the lesser repeated experiments that much more involved?  Were those repeated more frequently done to confirm more he experiment repeated 8 times (Figure 1, C) seemed to have a high standard deviation.  Could include other limitations if relevant. Experiments encompassing transduction of the HTLV-1-infected T-cell lines were more involved, which was burdensome as some shRNA plasmids (we tested several) did not work (e.g., Nrp1 not knocked down; off-target effects). Comparing RNA from empty vector- and HBZ-expressing cells, some genes were subject to more qRT-PCR runs because they served as positive controls for runs testing other genes. GEM was substantially up in HBZ cells; the error bar is due to a 25 to 436-fold increase (large error bar). We added a short statement in the Discussion regarding limitations, which centers on our inability to characterize cell-free virions produced from cells expressing or lacking Nrp1 (line 627).

Also, consider adding a conclusion and possible future directions for therapy options that this discovery could lead to? In the Discussion we state the possibility of using Nrp1 as a biomarker to gauge the risk of disease development (line 639). In addition, we state the possibility that small molecule inhibitors of Nrp1 could be a future therapy option for HTLV-1 patients (line 652).

Reviewer 2 Report

Comments to the Author

In this study, the authors analyzed the mechanism of Nrp1 expression and the inhibition of HTLV-1 infection by Nrp1. First, they showed that HBZ induced NRP1 transcription and increased c-Jun and JunB recruitment downstream of the NRP1 gene; p300 and CBP were recruited to the region. Next, the authors showed that overexpression of Nrp1 reduced the efficiency of HTLV-1 infection; the ectodomain of Nrp1 was responsible for the inhibition of HTLV-1 infection. This is an interesting report suggesting that activation of Nrp1 negatively impacts viral infection, while HBZ has been found to enhance HTLV-1 infection using cell-based models, as described by the authors.

Major comment

1.      In this report, the authors argue that approximately 50kb downstream of NRP1 is an enhancer for NRP1 transcription. However, the authors have not evaluated whether the sequence region they have identified functions as an enhancer. If the authors would like to describe it as an enhancer rather than a candidate enhancer, an experimental assay to verify enhancer function (Genomics. 2015, doi: 10.1016/j.ygeno.2015.06.005 ), such as enhancer luciferase reporter assay, is needed.

Author Response

We would like to thank all the reviewers for their positive feedback and constructive comments. To the best of our ability within this short revision period, we have attempted to answer their questions and address their comments, making some modifications to the manuscript. Our responses to the reviewers’ questions and comments (italicized) are below. Line numbers denoting where there are modifications to the manuscript correspond to the no mark-up version.

In this report, the authors argue that approximately 50kb downstream of NRP1 is an enhancer for NRP1 transcription. However, the authors have not evaluated whether the sequence region they have identified functions as an enhancer. If the authors would like to describe it as an enhancer rather than a candidate enhancer, an experimental assay to verify enhancer function (Genomics. 2015, doi: 10.1016/j.ygeno.2015.06.005 ), such as enhancer luciferase reporter assay, is needed. This point is certainly important. We based our language on its annotation as an enhancer in the UCSC Genome Browser according to location, ChIP-binding signatures and DNase I hypersensitivity; however, as we did not test this sequence in a luciferase assay, we have now emphasized that this enhancer should be denoted as a candidate enhancer (line 340).

Reviewer 3 Report

In this study, Kendle et al. found that the retroviral protein HBZ upregulates the transcription of genes involved in viral infection as well as NRP1, which encodes for the neuropilin 1 protein (Nrp1), that as been identified as a co-receptor for HTLV-1. In this study, they focused on the role of Nrp1 in HTLV-1 infection and they provided convincing results supporting a model in which HBZ up-regulates NRP1 transcription by augmenting recruitment of Jun proteins to an enhancer downstream of the gene. Very surprisingly they showed that Nrp1 overexpression on HTLV-1-infected cells inhibits viral infectivity of newly produced viral particles. Nrp1 was found to be incorporated into HTLV-1 virions, and the deletion of its ectodomain removed the inhibitory effect. These results proposed a new role for the HBZ protein in controlling the infectivity of HTLV-1 and establishing a chronic infection that could participate in the pathogenesis of the virus. This study is very interesting and well-conducted.

 Only a few minors’ points:

1-      It would be more convincing if the authors could show in Figure 1 an upregulation of HBZ at the surface membrane either by FACS analysis of their HeLa cells or by confocal microscopy following an immunostaining of NRP-1 and HTLV-1 env.

2-      Why the newly made HTLV-1 virus with more NRP-1 are less infectious? Do they contain fewer env proteins or does the overexpression of NRP-1 affect the virus's maturation or morphology? It would be very interesting if they could provide a WB of env with quantification and normalization of the western blot band relative intensity (with HTLV-1 Gap p19)

 3-      Usually, HTLV-1 is mainly transmitted by cell-cell contact, and in this study the authors used free virus particles. Do they observe the same effect of NRP-1 if they conduct infections through co-culture and spinoculation?

Author Response

We would like to thank all the reviewers for their positive feedback and constructive comments. To the best of our ability within this short revision period, we have attempted to answer their questions and address their comments, making some modifications to the manuscript. Our responses to the reviewers’ questions and comments (italicized) are below. Line numbers denoting where there are modifications to the manuscript correspond to the no mark-up version.

It would be more convincing if the authors could show in Figure 1 an upregulation of HBZ at the surface membrane either by FACS analysis of their HeLa cells or by confocal microscopy following an immunostaining of NRP-1 and HTLV-1 env. For us, the HeLa cell clones have served as a useful model system to examine how HBZ regulates transcription. However, following transfection with an HTLV-1 molecular clone or the single-cycle replication-dependent luciferase assay plasmids, these cells are unable to infect other cells in coculture. Therefore, they are not a good model for analyzing aspects of HTLV-1 infection (at least the HeLa cells that we have: HeLa-S3). We thought it was more relevant to confirm that Nrp1 is expressed on the surface of Jurkat and HTLV-1-infected T-cells.

Why the newly made HTLV-1 virus with more NRP-1 are less infectious? Do they contain fewer env proteins or does the overexpression of NRP-1 affect the virus's maturation or morphology? It would be very interesting if they could provide a WB of env with quantification and normalization of the western blot band relative intensity (with HTLV-1 Gap p19). We were interested in addressing this issue. However, following knockdown of Nrp1 in HTLV-1-infected T-cells, we did not obtain enough cells for subsequent isolation of cell-free virus. Furthermore, we were unable to obtain infectious virus or virus-like particles from HEK293T cells transfected with an HTLV-1 molecular clone or the single-cycle replication-dependent luciferase assay plasmids, respectively. This limitation of the study is addressed in a statement added to the Discussion (line 627).

Usually, HTLV-1 is mainly transmitted by cell-cell contact, and in this study the authors used free virus particles. Do they observe the same effect of NRP-1 if they conduct infections through co-culture and spinoculation? The infection assays shown in this study were done by coculture (cell-cell contact, clarified line 524). As mentioned above, it would be interesting to analyze how Nrp1 affects infection by cell-free virus, but we lacked an adequate experimental system to address this issue.

Reviewer 4 Report

In this manuscript, Kendle and Hoang et al. identify transcriptionally induced target genes of the HTLV-1-encoded HBZ, which was found to enhance HTLV-1 infection earlier. Specifically, the authors find upregulation of COL4A1, GEM, and NRP1 transcripts in HBZ-expressing cells. Interestingly, COL4A1 and GEM have been described to play a role in HTLV-1 infection and to be regulated by Tax, while NRP1 is part of the HTLV-1 receptor. The authors focus on NRP1 and find that HBZ increases recruitment of Jun family members to an enhancer downstream of the NRP1 gene. Moreover, they show enhanced recruitment of p300/CBP to the enhancer in presence of HBZ, thus, providing insights into the detailed HBZ-mediated transcriptional regulation of NRP1. In co-culture assays with Tax-responsive reporter cells, the authors find that repression of NRP1 increases HTLV-1 infection, while overexpression of NRP1 represses infection. Interestingly, neither Gag p19 nor gp46 Env expression are affected by manipulation of NRP1 expression levels. Mechanistically, the authors find that NRP1 is incorporated into viral particles, which may interfere with virion-target cell interaction, thus, reducing viral transmission. Overall, this is an interesting and well-written manuscript with technically sound experiments.

Minor comments:

Figure 1: Experiments in A,C,D were performed with HeLa clonal cell lines expressing HBZ, while in B, cells were transduced with a lentiviral vector. Why didn’t the authors use the same clonal cell line for the qPCR in subpanel B as well?

Fig. 1 E seems to show data from two different blots, please provide housekeeping gene for each blot.

Authors show transcriptional induction of COL4A1, COL4A2, and GEM. Does HBZ also induce protein expression of these target genes?

Figure 2: Some HTLV-1-infected cell lines do not express NRP1 (SLB-1). Authors should discus this.

Fig. 2B seems to show data from two different blots, please provide housekeeping gene for each blot. Please check also Fig. 6B.

Figure 5: The Western Blot of Fig. 5I showing repression of NRP1 and expression of viral proteins in Jurkat cells is missing.

Author Response

We would like to thank all the reviewers for their positive feedback and constructive comments. To the best of our ability within this short revision period, we have attempted to answer their questions and address their comments, making some modifications to the manuscript. Our responses to the reviewers’ questions and comments (italicized) are below. Line numbers denoting where there are modifications to the manuscript correspond to the no mark-up version.

Figure 1: Experiments in A,C,D were performed with HeLa clonal cell lines expressing HBZ, while in B, cells were transduced with a lentiviral vector. Why didn’t the authors use the same clonal cell line for the qPCR in subpanel B as well? We performed qRT-PCR on specimens from the HeLa clonal cells lines and replaced these new data with the transduced cells data.

Fig. 1 E seems to show data from two different blots, please provide housekeeping gene for each blot. We are now showing HBZ and beta-actin on the same blot.

Authors show transcriptional induction of COL4A1, COL4A2, and GEM. Does HBZ also induce protein expression of these target genes? In western blots, we did not detect a significant difference in COL4 protein levels between the empty vector and HBZ-expressing HeLa clones. However, the COL4 antibody (ab6586, Abcam) produced an intense signal, which led us to suspect that the already high abundance of the protein in the absence of HBZ might mask an effect on expression by HBZ. The GEM antibody we purchased did not work in our western blot assays, and we did not order another antibody to pursue this line of inquiry.

Figure 2: Some HTLV-1-infected cell lines do not express NRP1 (SLB-1). Authors should discuss this. This is an important point that we did try to address in the Discussion (paragraph starting on line 598). To this paragraph, we now discuss results and speculation regarding the SLB-1 cells.

Fig. 2B seems to show data from two different blots, please provide housekeeping gene for each blot. Please check also Fig. 6B. The beta-actin blots have been added.

Figure 5: The Western Blot of Fig. 5I showing repression of NRP1 and expression of viral proteins in Jurkat cells is missing. We are limited in the ability to see repression of Nrp1 and detect viral proteins by western blot with this system. For us, electroporation of Jurkat cells results in a low percentage of cells with transgene expression. We use 15-20 micrograms of total plasmid per electroporation. While we might detect transgene protein expression with this quantity of a single plasmid, in the experiment shown in Fig. 5I, we are electroporating cells with four plasmids. Nrp1 knock-down was confirmed in ATL-2 cells (Fig. 5F) and HeLa cells transfected with knockdown vector (data not shown).

Round 2

Reviewer 2 Report

The authors sufficiently addressed the reviewer's concerns. I have no additional comments.